# Optimal Control of Background-Based Uncertain Systems with Applications in DC Pension Plan

**DOI:** 10.3390/e24050734

**Published:** 2022-05-21

**Authors:** Wei Liu, Wanying Wu, Xiaoyi Tang, Yijun Hu

**Affiliations:** 1College of Mathematics and System Science, Xinjiang University, Urumqi 830046, China; liuwei.math@xju.edu.cn (W.L.); xytang_math@stu.xju.edu.cn (X.T.); 2School of Mathematics and Statistics, Wuhan University, Wuhan 430072, China; yjhu.math@whu.edu.cn

**Keywords:** optimal control, uncertainty theory, optimistic value, defined contribution pension plan, background-state variable

## Abstract

In this paper, we propose a new optimal control model for uncertain systems with jump. In the model, the background-state variables are incorporated, where the background-state variables are governed by an uncertain differential equation. Meanwhile, the state variables are governed by another uncertain differential equation with jump, in which both the background-state variables and the control variables are involved. Under the optimistic value criterion, using uncertain dynamic programming method, we establish the principle and the equation of optimality. As an application, the optimal investment strategy and optimal payment rate for DC pension plans are given, where the corresponding background-state variables represent the salary process. This application in DC pension plans illustrates the effectiveness of the proposed model.

## 1. Introduction

Since [1] first discussed the problem of stochastic control with jump, the problem of stochastic optimal control with jump has become an important branch of control theory. From the perspectives of theory and applications, especially including in finance and insurance, the involved stochastic differential equations with and without jump have been extensively studied. For example, see [2,3,4,5,6,7,8,9,10,11,12,13,14,15,16,17,18,19,20], and the references therein.

The above-mentioned references are based on probability models. However, in reality, the finance markets usually are of model uncertainty, which means that it is difficult to determine the specific probability. Therefore, it is of great importance to study uncertainty theory with its applications in finance and insurance. For general theory and applications about uncertainty theory and optimal control of uncertain systems, we refer to [21,22,23,24,25,26,27,28,29,30,31] and the references therein. For the applications of uncertainty theory in option pricing theory and portfolio selections, see [26,27,28,29,32,33,34,35], and the references therein. For the applications of uncertainty theory in insurance, especially in pension plans, see [22,36]. In the study of optimal control of uncertain systems, the optimality criteria mainly include four criteria: expected value criterion, optimistic value criterion, pessimistic value criterion, and Hurwicz criterion. Under the optimistic value criterion, the principle of optimality and the equation of optimality for uncertain systems without jump were discussed by [26]. Recently, Ref. [27] studied the optimal control of uncertain systems with jump under the optimistic value criterion, where the state variables are governed by an uncertain differential equation with jump. Later, Ref. [37] extended those of [27] to the multidimensional setting. Nevertheless, from both the theoretical and practical point of view, the state variables are usually also affected by the environment factors except the control variables. Therefore, it is interesting and necessary to consider the optimal control of uncertain systems by incorporating the environment factors into the optimal control models under the optimistic value criterion.

In this paper, we propose a new optimal control model for uncertain systems under the optimistic value criterion. Namely, the environment factors are first understood as background variables. Then we assume that the background-state variables are governed by an uncertain differential equation, and further we assume that the state variables are also governed by another uncertain differential equation with jump in which both the background-state variable and the control variables are involved. By making use of the uncertain dynamic programming method, both the principle and the equation of optimality are established. Finally, as an application, the optimal investment strategy and the optimal payment rate for DC pension plans are discussed, where the corresponding background-state variables represent the salary process. This application in DC pension plans illustrates the effectiveness of the proposed model.

The rest of the paper is organized as follows. In Section 2, we introduce preliminaries, including basic notations of uncertainty theory. In Section 3, the optimistic value models for uncertain systems with jump are introduced and the principle of optimality is provided. Section 4 is devoted to the equation of optimality. In Section 5, as an application of the proposed optimal control model in DC pension plans, the optimal investment strategy and the optimal payment rate are obtained. Section 6 presents numerical analysis to illustrate our results. Finally, the conclusions are summarized.

## 2. Preliminary

### 2.1. Uncertainty Space

In this subsection, we collect some basic definitions of uncertainty theory which are from [21,23,24].

Let Γ be a nonempty set, and L a σ-algebra over Γ. Each element A∈L is called an event. A set function M defined on the σ-algebra L over Γ is called an uncertain measure if it satisfies the following four conditions:(i)M{Γ}=1,(ii)$M\left\{\Lambda \right\}+M\left\{\Lambda^{c}\right\}=1$ for any event A∈L,(iii)$M\left\{{⋃}_{i=1}^{\infty }\Lambda_{i}\right\}\leq \sum_{i=1}^{\infty }M\left\{\Lambda_{i}\right\}$ for every countable sequence of events $\Lambda_{i}$.(iv)Let $\left(\Gamma_{k},L_{k},M_{k}\right)$ be uncertainty spaces for k=1,2,⋯n. The product uncertain measure is
$$M\left\{\prod_{i=1}^{n}\Lambda_{k}\right\}=\min_{1\leq i\leq n}M\left\{\Lambda_{k}\right\}.$$

**Definition** **1.**
*Let *Γ* be a nonempty set, L a σ-algebra over *Γ* and M an uncertain measure. Then the triplet (Γ,L,M) is called an uncertainty space.*


**Definition** **2.**
*An uncertain variable is a function ξ from an uncertainty space (Γ,L,M) to the set of real numbers such that {ξ∈B}:={γ∈Γ∣ξ(γ)∈B} s an event for any Borel set B.*


**Definition** **3.**
*The uncertainty distribution Φ:R→[0,1] of an uncertain variable ξ is defined by*

Φ(x):=M{ξ≤x},x∈R.



The following lemma is a characterization of an uncertainty distribution, which is from [25].

**Lemma** **1.**
*A function Φ(x):R→[0,1] is an uncertainty distribution if and only if it is a monotone increasing function except Φ(x)=0 and Φ(x)=1.*


**Definition** **4.**
*Let ξ be an uncertain variable. Then the expected value of ξ is defined by*

$$E\left[\xi \right]:=\int_{0}^{+\infty }M\left\{\xi \geq x\right\}dx-\int_{-\infty }^{0}M\left\{\xi \leq x\right\}dx,$$

*provided that at least one of the two integrals is finite.*


**Definition** **5.**
*Let ξ be an uncertain variable with finite expected value E[ξ]. Then the variance of ξ is defined by*

$$V\left[\xi \right]:=E[{\left(\xi -E\left[\xi \right]\right)}^{2}].$$



**Definition** **6.**
*The uncertain variables $\xi_{1},\xi_{2},\ldots ,\xi_{n}$ are said to be independent if*

$$M\left\{\overset{n}{\underset{i=1}{⋂}}\left(\xi_{i}\in B_{i}\right)\right\}=\min_{1\leq i\leq n}M\left\{\xi_{i}\in B_{i}\right\},$$

*for any Borel sets $B_{1},B_{2},\cdot \cdot \cdot ,B_{n}$ of real numbers.*


### 2.2. Optimistic Value and Pessimistic Value

**Definition** **7.**
*Let ξ be an uncertain variable, and α∈(0,1]. Then $\xi_{\sup}\left(\alpha \right):=\sup\left\{r|M\left\{\xi \geq r\right\}\geq \alpha \right\}$ is called the α-optimistic value to ξ; and $\xi_{\inf}\left(\alpha \right):=\inf\left\{r|M\left\{\xi \leq r\right\}\geq \alpha \right\}$ is called the α-pessimistic value to ξ.*


The following lemma is about properties of optimistic value, which is from [21,24].

**Lemma** **2.**
*Assume that ξ and η are independent uncertain variables and α∈(0,1]. Then we have*
*(i)* 
*if λ≥0, then ${\left(\lambda \xi \right)}_{\sup}\left(\alpha \right)=\lambda \xi_{\sup}\left(\alpha \right)$, and ${\left(\lambda \xi \right)}_{\inf}\left(\alpha \right)=\lambda \xi_{\inf}\left(\alpha \right)$;*
*(ii)* 
*λ<0, then ${\left(\lambda \xi \right)}_{\sup}\left(\alpha \right)=\lambda \xi_{\inf}\left(\alpha \right)$, and ${\left(\lambda \xi \right)}_{\inf}\left(\alpha \right)=\lambda \xi_{\sup}\left(\alpha \right)$;*
*(iii)* 
*${\left(\xi +\eta \right)}_{\sup}\left(\alpha \right)=\xi_{\sup}\left(\alpha \right)+\eta_{\sup}\left(\alpha \right)$, ${\left(\xi +\eta \right)}_{\inf}\left(\alpha \right)=\xi_{\inf}\left(\alpha \right)+\eta_{\inf}\left(\alpha \right)$.*



**Definition** **8.**
*An uncertain process $C_{t}$ is said to be a Liu process if*
*(i)* 
*$C_{0}=0$ and almost all sample paths are Lipschitz continuous,*
*(ii)* 
*$C_{t}$ has stationary and independent increments,*
*(iii)* 
*every increment $C_{s+t}-C_{s}$ is a normal distributed uncertain variable with expected value 0 and variance $t^{2}$, whose uncertainty distribution is*

$$\Phi \left(x\right):={\left(1+\exp\left(\frac{-\pi x}{\sqrt{3}t}\right)\right)}^{-1},\;x\in R.$$




Let $C_{t}$ be a Liu process, and $\xi :=\Delta C_{t}:=C_{t+\Delta t}-C_{t}$. Then for any 0<α<1, α-optimistic and α-pessimistic values of ξ are
(1)$$\xi_{\sup}\left(\alpha \right)=\frac{\sqrt{3}\Delta t}{\pi }\ln\frac{1-\alpha }{\alpha }$$
and
(2)$$\xi_{\inf}\left(\alpha \right)=-\frac{\sqrt{3}\Delta t}{\pi }\ln\frac{1-\alpha }{\alpha },$$
respectively, for example, see Example 1.7 of [30] or (1) and (2) of [27].

The following definitions are about optimal control with jump of uncertainty theory, which are from [38].

**Definition** **9.**
*An uncertain process $V_{t}$ is said to be a V jump process with parametrs $r_{1}$ and $r_{2}$$\left(0<r_{1}<r_{2}<1\right)$ for t≥0 if*
*(i)* 
*$V_{0}=0$,*
*(ii)* 
*$V_{t}$ has stationary and independent increments,*
*(iii)* 
*every increment $V_{s+t}-V_{s}$ is a Z jump uncertain variable $Z(r_{1},r_{2},t)$, whose uncertainty distribution is*

(3)
$$\Psi \left(x\right):=\left\{\begin{array}{cc} \;0, & \mathrm{if}\;x<0,\\ \;\frac{2r_{1}}{t}x, & \mathrm{if}\;0\leq x\leq \frac{t}{2},\\ \;r_{2}+\frac{2(1-r_{2})}{t}\left.(x-\frac{t}{2}\right.), & \mathrm{if}\;\frac{t}{2}\leq x<t,\\ \;1, & \mathrm{if}\;x\geq t. \end{array}\right.$$




Let $V_{t}$ be a *V* jump uncertain process, and $\eta =\Delta V_{t}=V_{t+\Delta t}-V_{t}$. Then for any α∈(0,1), it follows from the definition of α-optimistic value and α-pessimistic value that
(4)$$\eta_{\sup}\left(\alpha \right)=\left\{\begin{array}{cc} \;\left.(1-\frac{\alpha }{2(1-r_{2})}\right.)\Delta t, & \mathrm{if}\;0<\alpha <1-r_{2},\\ \;\frac{\Delta t}{2}, & \mathrm{if}\;1-r_{2}\leq \alpha <1-r_{1},\\ \;\frac{1-\alpha }{2r_{1}}\Delta t, & \mathrm{if}\;1-r_{1}\leq \alpha <1, \end{array}\right.$$
and
(5)$$\eta_{\inf}\left(\alpha \right)=\left\{\begin{array}{cc} \;\frac{\alpha }{2r_{1}}\Delta t, & \mathrm{if}\;0<\alpha \leq r_{1},\\ \;\frac{\Delta t}{2}, & \mathrm{if}\;r_{1}<\alpha \leq r_{2},\\ \;\left.(1-\frac{1-\alpha }{2(1-r_{2})}\right.)\Delta t, & \mathrm{if}\;r_{2}<\alpha <1, \end{array}\right.$$
respectively.

**Definition** **10.**
*Suppose that $C_{t}$ is an Liu process, f and g are two functions. Then*

(6)
$$dX_{t}=f\left(t,X_{t}\right)dt+g\left(t,X_{t}\right)dC_{t}$$

*is called an uncertain differential equation. A solution is a uncertain process $X_{t}$ that satisfies (Equation 6) identically in t.*


**Remark** **1.**
*The uncertain differential Equation (Equation 6) means the solution $X_{t}$ meets the uncertain integral equation*

$$X_{t}=X_{0}+\int_{0}^{t}f\left(t,X_{t}\right)ds+\int_{0}^{t}g\left(t,X_{t}\right)dC_{s}.$$



**Definition** **11.**
*Suppose that $C_{t}$ is a Liu process, $V_{t}$ is an uncertain V jump process, and $g_{1}$, $g_{2}$ and $g_{3}$ are some given functions. Then*

(7)
$$dX_{t}=g_{1}\left(t,X_{t}\right)dt+g_{2}(t,\left(X_{t}\right)dC_{t}+g_{3}\left(t,X_{t}\right)dV_{t}$$

*is called an uncertain differential equation with jump. A solution is an uncertain process $X_{t}$ that satisfies (Equation 7) identically in t.*


## 3. Optimistic Value Model under Background-State of Uncertain Optimal Control with Jump

In the problem of uncertain optimal control, we should determine some optimization criteria to optimize objective function of involving uncertain variables and convert the uncertain objective to its definite equivalent goal. In the uncertain optimal control, there are many criteria, for example: expected value, optimistic value, pessimistic value and Hurwicz criterion. Under [27], they discussed optimal control problem of uncertain dynamic systems with jump under the optimistic value criterion. In this paper, we involve the background-state variables and discuss optimal control problem under the optimistic value criterion for this kind of systems, where both the background-state variables and the control variables are involved.

Assume that $C_{t}$ is a Liu process, $V_{t}$ is an uncertain *V*-jump process with parameters $r_{1}$ and $r_{2}$$\left(0<r_{1}<r_{2}<1\right)$, where $C_{t}$ and $V_{t}$ are independent. The confidence level α∈(0,1). For any 0<t<T, an optimistic value model of uncertain optimal control problem with jump as follows
(8)$$\left\{\begin{array}{c} \begin{array}{cc} \; & J\left(t,x,l\right)=\underset{D_{t}\in D}{\sup}{\left[\int_{t}^{T}f\left(s,X_{s},L_{s},D_{s}\right)ds+G\left(T,X_{T},L_{T}\right)\right]}_{\sup}\left(\alpha \right)\;\\ \; & \mathrm{subject}\;\mathrm{to}\;\\ \; & dX_{s}=\nu \left(s,X_{s},L_{s},D_{s}\right)ds+\lambda \left(s,X_{s},L_{s},D_{s}\right)dC_{s}+\chi \left(s,X_{s},L_{s},D_{s}\right)dV_{s}\;,\\ \; & dL_{s}=m\left(L_{s}\right)ds+n\left(L_{s}\right)dC_{s},\;X_{t}=x,\;L_{t}=l, \end{array} \end{array}\right.$$
where $X_{s}$ is the state variable, $L_{s}$ is called the background-state variable, $D_{s}$ is the control variable and it subject to a constraint set D. The function f:[0,T]×R×R×R→R is the objective function, and the function G:[0,T]×R×R→R is the terminal reward. ${\left[\int_{t}^{T}f\left(s,X_{s},L_{s},D_{s}\right)ds+G\left(T,X_{T},L_{T}\right)\right]}_{\sup}\left(\alpha \right)$ denotes the α-optimistic value to the uncertain variable in middle bracket. ν, λ, χ are three functions of time *s*, state $X_{s}$, background-state $L_{s}$ and control $D_{s}$. Furthermore, *m*, *n* are two functions of $L_{s}$. All functions mentioned are continuous. Now we give the following principle of optimality to solve the proposed model.

**Theorem** **1.**
*For any (t,x,l)∈[0,T)×R×R, Δt>0 with t+Δt<T, we have*

(9)
$$J\left(t,x,l\right)=\underset{D_{t}\in D}{\sup}{\left\{f\left(t,x,l,D_{t}\right)\Delta t+J\left(t+\Delta t,x+\Delta X_{t},l+\Delta L_{t}\right)+o\left(\Delta t\right)\right\}}_{\sup}\left(\alpha \right),$$

*where $x+\Delta X_{t}=X_{t+\Delta t},l+\Delta L_{t}=L_{t+\Delta t}$.*


**Proof.** We use $\bar{J}\left(t,x,l\right)$ to denote the right side of (Equation 9). It follows from the definition of J(t,x,l) that
$$\begin{array}{cc} J(t,x,l)\geq & \{\int_{t}^{t+\Delta t}f\left(s,X_{s},L_{s},D_{s}{|}_{\left[t,t+\Delta t\right)}\right)ds\\ & +\int_{t+\Delta t}^{T}f\left(s,X_{s},L_{s},D_{s}{|}_{\left[t+\Delta t,T\right]}\right)ds+G\left(T,X_{T},L_{T}\right){\}}_{\sup}\left(\alpha \right), \end{array}$$
for any $D_{s}$, where $D_{s}{|}_{\left[t,t+\Delta t\right)}$ and $D_{s}{|}_{\left[t+\Delta t,T\right]}$ are the values of decision variable $D_{s}$ restricted on [t,t+Δt) and [t+Δt,T], respectively.For any Δt>0, by using Taylor series expansion, we get
$$\int_{t}^{t+\Delta t}f\left(s,X_{s},L_{s},D_{s}{|}_{\left[t,t+\Delta t\right)}\right)ds=f\left(t,x,l,D\left(t,x,l\right)\right)\Delta t+o\left(\Delta t\right).$$Thus
(10)$$\begin{array}{cc} J(t,x,l)\geq & f\left(t,x,l,D_{t}\right)\Delta t+o\left(\Delta t\right)\\ & +{\left\{\int_{t+\Delta t}^{T}f\left(s,X_{s},L_{s},D_{s}{|}_{\left[t+\Delta t,T\right]}\right)ds+G\left(T,X_{T},L_{T}\right)\right\}}_{\sup}\left(\alpha \right). \end{array}$$Taking the supremum with respect to $D_{s}{|}_{\left[t+\Delta t,T\right]}$ in (Equation 10), then we get
$$\underset{D_{t}\in D}{\sup}{\left\{\int_{t+\Delta t}^{T}f\left(s,X_{s},L_{s},D_{s}{|}_{\left[t+\Delta t,T\right]}\right)ds+G\left(T,X_{T},L_{T}\right)\right\}}_{\sup}\left(\alpha \right)=J\left(t+\Delta t,x+\Delta X_{t},l+\Delta L_{t}\right).$$Then the right side of (Equation 10) becomes that
$$\underset{D_{t}\in D}{\sup}{\left\{f\left(t,x,l,D_{t}\right)\Delta t+J\left(t+\Delta t,x+\Delta X_{t},l+\Delta L_{t}\right)+o\left(\Delta t\right)\right\}}_{\sup}\left(\alpha \right)=\bar{J}\left(t,x,l\right).$$
Now we get $J\left(t,x,l\right)\geq \bar{J}\left(t,x,l\right)$.On the other hand, for all $D_{s}$, we have
$$\begin{array}{cc} \; & {\left\{\int_{t}^{T}f\left(s,X_{s},L_{s},D_{s}\right)ds+G\left(T,X_{T},L_{T}\right)\right\}}_{\sup}\left(\alpha \right)\\ & =f\left(t,x,l,D_{t}\right)\Delta t+o\left(\Delta t\right)+{\left\{\int_{t+\Delta t}^{T}f\left(s,X_{s},L_{s},D_{s}{|}_{\left[t+\Delta t,T\right]}\right)ds+G\left(T,X_{T},L_{T}\right)\right\}}_{\sup}\left(\alpha \right)\\ & \leq f\left(t,x,l,D_{t}\right)\Delta t+o\left(\Delta t\right)+\left[J\left(t+\Delta t,x+\Delta X_{t},l+\Delta L_{t}\right)\right]\\ & \leq \bar{J}\left(t,x,l\right). \end{array}$$
Hence, $J\left(t,x,l\right)\leq \bar{J}\left(t,x,l\right)$, and then $J\left(t,x,l\right)=\bar{J}\left(t,x,l\right)$. Theorem 1 is proved. □

## 4. Optimality Condition

We derive the following equation of optimality by the principle of optimality above.

**Theorem** **2.**
*Let J(t,x,l) be twice continuously differentiable on [0,T]×R×R. Then we have*

(11)
$$-J_{t}=\underset{D_{t}\in D}{\sup}\left\{f+\nu J_{x}+m\left(L_{t}\right)J_{l}+\frac{\sqrt{3}}{\pi }\ln\frac{1-\alpha }{\alpha }\left|(\lambda J_{x}+n\left(L_{t}\right)J_{l})\right|+k\left|\chi J_{x}\right|\right\},$$

*where $J_{t}$, $J_{x}$ and $J_{l}$ are the partial derivatives of the function J(t,x,l) in t, x and l, respectively, and f, ν, λ, χ, $J_{t}$, $J_{x}$, $J_{l}$, $J_{tt}$, $J_{xx}$, $J_{ll}$, $J_{tx}$, $J_{tl}$, $J_{xl}$ denote $f(t,x,l,D_{t})$, $\nu (t,x,l,D_{t})$, $\lambda (t,x,l,D_{t})$, $\chi (t,x,l,D_{t})$, $J_{t}\left(t,x,l\right)$, $J_{x}\left(t,x,l\right)$, $J_{l}\left(t,x,l\right)$, $J_{tt}\left(t,x,l\right)$, $J_{xx}\left(t,x,l\right)$, $J_{ll}\left(t,x,l\right)$, $J_{tx}\left(t,x,l\right)$, $J_{tl}\left(t,x,l\right)$, $J_{xl}\left(t,x,l\right)$, respectively, and*
*(1)* 
*when $\chi J_{x}\geq 0$, (i) if $0<\alpha <1-r_{2}$, then $k=1-\frac{\alpha }{2(1-r_{2})}$, (ii) if $1-r_{2}\leq \alpha <1-r_{1}$, then $k=\frac{1}{2}$, (iii) if $1-r_{1}\leq \alpha <1$, then $k=\frac{1-\alpha }{2r_{1}}$;*
*(2)* 
*when $\chi J_{x}<0$, (i) if $0<\alpha <r_{1}$, then $k=\frac{\alpha }{2r_{1}}$, (ii) if $r_{1}<\alpha \leq r_{2}$, then $k=\frac{1}{2}$, (iii) if $r_{2}<\alpha <1$, then $k=1-\frac{1-\alpha }{2(1-r_{2})}$.*



**Remark** **2.**
*Suppose state variable $X_{t}$ does not depend on $L_{t}$, the J in (Equation 8) is a function of (t,x), therefore Theorem 2 goes back to the classic setting, for example, see the model in [37].*


**Proof.** For any Δt>0, note that there exist constants $K_{1}>0$ and $K_{2}>0$, such that $\Delta X_{t}\leq K_{1}\Delta t$ and $\Delta L_{t}\leq K_{2}\Delta t$, by Taylor series expansion, we have that
(12)$$\begin{array}{cc} J(t+\Delta t,x+\Delta X_{t},l+\Delta L_{t}) & =J\left(t,x,l\right)+J_{t}\left(t,x,l\right)\Delta t+J_{x}\left(t,x,l\right)\Delta X_{t}+J_{l}\left(t,x,l\right)\Delta L_{t}\\ \; & +\frac{1}{2}J_{tt}\left(t,x,l\right)\Delta t^{2}+\frac{1}{2}J_{xx}\left(t,x,l\right)\Delta X_{t}^{2}+\frac{1}{2}J_{ll}\left(t,x,l\right)\Delta L_{t}^{2}\\ \; & +J_{tx}\left(t,x,l\right)\Delta t\Delta X_{t}+J_{tl}\left(t,x,l\right)\Delta t\Delta L_{t}+J_{xl}\left(t,x,l\right)\Delta X_{t}\Delta L_{t}\\ \; & +o(\Delta t^{2}), \end{array}$$
where the infinitesimal $o(\Delta t^{2})$ satisfies that
$$\lim_{\Delta t\to 0}\frac{o(\Delta t^{2})}{\Delta t^{2}}=0.$$
Note that $\Delta X_{t}=\nu \left(t,X_{t},L_{t},D_{t}\right)\Delta t+\lambda \left(t,X_{t},L_{t},D_{t}\right)\Delta C_{t}+\chi \left(t,X_{t},L_{t},D_{t}\right)\Delta V_{t}$, $\Delta L_{t}=m\left(L_{t}\right)\Delta t+n\left(L_{t}\right)\Delta C_{t}$. Substituting (Equation 12) into (Equation 9) yields that
(13)$$\begin{array}{cc} J\left(t,x,l\right)=\underset{D_{t}\in D}{\sup} & \{f\left(t,x,l,D_{t}\right)\Delta t+[J\left(t,x,l\right)+J_{t}\left(t,x,l\right)\Delta t+J_{x}\left(t,x,l\right)\Delta X_{t}\\ \; & +\frac{1}{2}J_{tt}\left(t,x,l\right)\Delta t^{2}+\frac{1}{2}J_{xx}\left(t,x,l\right)\Delta X_{t}^{2}+\frac{1}{2}J_{ll}\left(t,x,l\right)\Delta L_{t}^{2}\\ \; & +J_{l}\left(t,x,l\right)\Delta L_{t}+J_{tx}\left(t,x,l\right)\Delta t\Delta X_{t}+J_{tl}\left(t,x,l\right)\Delta t\Delta L_{t}\\ \; & +J_{xl}\left(t,x,l\right)\Delta X_{t}\Delta L_{t}+o\left(\Delta t^{2}\right){]}_{\sup}\left(\alpha \right)\}. \end{array}$$Then we have
(14)$$\begin{array}{cc} 0=\underset{D_{t}\in D}{\sup} & \{f\left(t,x,l,D_{t}\right)\Delta t+J_{t}\left(t,x,l\right)\Delta t+[J_{x}\left(t,x,l\right)\Delta X_{t}+J_{l}\left(t,x,l\right)\Delta L_{t}\\ \; & +\frac{1}{2}J_{tt}\left(t,x,l\right)\Delta t^{2}+\frac{1}{2}J_{xx}\left(t,x,l\right)\Delta X_{t}^{2}+\frac{1}{2}J_{ll}\left(t,x,l\right)\Delta L_{t}^{2}\\ \; & +J_{tx}\left(t,x,l\right)\Delta t\Delta X_{t}+J_{tl}\left(t,x,l\right)\Delta t\Delta L_{t}+J_{xl}\left(t,x,l\right)\Delta X_{t}\Delta L_{t}{]}_{\sup}\left(\alpha \right)\\ \; & +o(\Delta t^{2})\}. \end{array}$$
Let uncertain variable $\xi =\Delta C_{t}$, $\eta =\Delta V_{t}$, then it follows from the uncertain differential equation in model (Equation 8), that
(15)$$\Delta X_{t}=\nu \Delta t+\lambda \xi +\chi \eta ,\;\Delta L_{t}=m\left(L_{t}\right)\Delta t+n\left(L_{t}\right)\xi ,$$
where ν, λ, χ denote $\nu (t,X_{t}$,$L_{t},D_{t})$, $\lambda (t,X_{t},L_{t},D_{t})$, $\chi (t,X_{t},L_{t},D_{t})$, respectively.Substituting (Equation 15) into (Equation 14) yields that
(16)$$\begin{array}{cc} 0= & \underset{D_{t}\in D}{\sup}\{f\left(t,x,l,D_{t}\right)\Delta t+J_{t}\Delta t+\nu J_{x}\Delta t+m\left(L_{t}\right)J_{l}\Delta t+[(\lambda J_{x}+n\left(L_{t}\right)J_{l}+\nu \lambda J_{xx}\Delta t\\ \; & +m\left(L_{t}\right)n\left(L_{t}\right)J_{ll}\Delta t+\lambda J_{tx}\Delta t+n\left(L_{t}\right)J_{tl}\Delta t+\nu n\left(L_{t}\right)J_{xl}\Delta t+\lambda m\left(L_{t}\right)J_{xl}\Delta t)\xi \\ \; & +\left(\frac{1}{2}\lambda^{2}J_{xx}+\frac{1}{2}n{\left(L_{t}\right)}^{2}J_{ll}+\lambda n\left(L_{t}\right)J_{xl}\right)\xi^{2}\\ \; & +\left(\chi J_{x}+\nu \chi J_{xx}\Delta t+\chi J_{tx}\Delta t+m\left(L_{t}\right)\chi J_{xl}\Delta t\right)\eta +\left(\frac{1}{2}\chi^{2}J_{xx}\right)\eta^{2}\\ \; & +\left(\lambda \chi J_{xx}+n\left(L_{t}\right)\chi J_{xl}\right)\xi \eta {]}_{\sup}\left(\alpha \right)+o\left(\Delta t^{2}\right)\}\\ \; & =\underset{D_{t}\in D}{\sup}\left\{f\Delta t+J_{t}\Delta t+\nu J_{x}\Delta t+m\left(L_{t}\right)J_{l}\Delta t+{\left[a\xi +b\xi^{2}+p\eta +q\eta^{2}+f\xi \eta \right]}_{\sup}\left(\alpha \right)+o\left(\Delta t^{2}\right)\right\}, \end{array}$$
where
$$\begin{array}{cc} a & =\lambda \left[J_{x}+\nu J_{xx}\Delta t+J_{tx}\Delta t+m\left(L_{t}\right)J_{xl}\Delta t\right]+n\left(L_{t}\right)\left[J_{l}+m\left(L_{t}\right)J_{ll}\Delta t+J_{tl}\Delta t+\nu J_{xl}\Delta t\right],\\ b & =\lambda^{2}\left(\frac{1}{2}J_{xx}\right)+\frac{1}{2}n{\left(L_{t}\right)}^{2}J_{ll}+\lambda n\left(L_{t}\right)J_{xl},\\ p & =\chi [J_{x}+\nu J_{xx}\Delta t+J_{tx}\Delta t+m\left(L_{t}\right)J_{xl}\Delta t],\\ q & =\chi^{2}\left(\frac{1}{2}J_{xx}\right),\\ f & =\lambda \chi J_{xx}+n\left(L_{t}\right)\chi J_{xl}. \end{array}$$
Let $Q\left(\xi ,\eta \right)=a\xi +b\xi^{2}+p\eta +q\eta^{2}+f\xi \eta$, thus the equation becomes
(17)$$0=\underset{D_{t}\in D}{\sup}\left\{f\Delta t+J_{t}\Delta t+\nu J_{x}\Delta t+m\left(L_{t}\right)J_{l}\Delta t+{\left[Q\right]}_{\sup}\left(\alpha \right)+o\left(\Delta t^{2}\right)\right\},$$
where ${\left[Q\left(\xi ,\eta \right)\right]}_{\sup}\left(\alpha \right)$ is denoted by ${\left[Q\right]}_{\sup}\left(\alpha \right)$.Since $|b\xi^{2}|\leq \frac{1}{2}\left|b\right|\left(\xi^{2}+\xi^{2}\right)$, $|q\eta^{2}|\leq \frac{1}{2}\left|q\right|\left(\eta^{2}+\eta^{2}\right)$, $\left|f\xi \eta \right|\leq \frac{1}{2}\left|f\right|\left(\xi^{2}+\eta^{2}\right)$, we have
(18)$$\begin{array}{cc} a\xi +p\eta -\left[\left(|b\right|+\frac{1}{2}\left|f|\right)\xi^{2}+(\frac{1}{2}\left|f\right|+\left|q|\right)\eta^{2}\right]\; & \leq a\xi +b\xi^{2}+p\eta +q\eta^{2}+f\xi \eta \\ \; & \leq a\xi +p\eta +\left[\left(|b\right|+\frac{1}{2}\left|f|\right)\xi^{2}+(\frac{1}{2}\left|f\right|+\left|q|\right)\eta^{2}\right] \end{array}$$It follows from the independence of ξ and η that
(19)$$\begin{array}{cc} {}[a\xi +p\eta & -\left\{\left(|b\right|+\frac{1}{2}\left|f|\right)\xi^{2}+(\frac{1}{2}\left|f\right|+\left|q|\right)\eta^{2}\right\}{]}_{\sup}\left(\alpha \right)\\ \; & \leq {\left[a\xi +b\xi^{2}+p\eta +q\eta^{2}+f\xi \eta \right]}_{\sup}\left(\alpha \right)\\ \; & \leq {\left[a\xi +p\eta +\left\{\left(|b\right|+\frac{1}{2}\left|f|\right)\xi^{2}+(\frac{1}{2}\left|f\right|+\left|q|\right)\eta^{2}\right\}\right]}_{\sup}\left(\alpha \right) \end{array}$$According to Theorem 4 in Sheng and Zhu (2013), for any ε>0 small enough, we have
(20)$$\begin{array}{c} {\left[a\xi +p\eta -\left(|b\right|+\frac{1}{2}\left|f|\right)\xi^{2}\right]}_{\sup}\left(\alpha \right)\geq \frac{\sqrt{3}}{\pi }\ln\frac{1-\alpha -\varepsilon }{\alpha +\varepsilon }|a|\Delta t-{\left(\frac{\sqrt{3}}{\pi }\ln\frac{2-\varepsilon }{\varepsilon }\right)}^{2}\left(|b\right|+\frac{1}{2}|f|)\Delta t^{2} \end{array}$$
(21)$$\begin{array}{c} {\left[a\xi +p\eta +\left(|b\right|+\frac{1}{2}\left|f|\right)\xi^{2}\right]}_{\sup}\left(\alpha \right)\leq \frac{\sqrt{3}}{\pi }\ln\frac{1-\alpha +\varepsilon }{\alpha -\varepsilon }|a|\Delta t+{\left(\frac{\sqrt{3}}{\pi }\ln\frac{2-\varepsilon }{\varepsilon }\right)}^{2}\left(|b\right|+\frac{1}{2}|f|)\Delta t^{2} \end{array}$$According to Theorem 5.1 in Deng et al. (2018), we have
(1)if p≥0, then
(22)$${\left[p\eta -\left(|q\right|+\frac{1}{2}\left|f|\right)\eta^{2}\right]}_{\sup}\left(\alpha \right)\geq \left\{\begin{array}{cc} \left(1-\frac{\alpha }{2(1-r_{2})}\right)p\Delta t-\left(|q\right|+\frac{1}{2}|f|)\Delta t^{2}, & \mathrm{if}\;0<\alpha <1-r_{2},\\ \frac{1}{2}p\Delta t-\left(|q\right|+\frac{1}{2}|f|)\Delta t^{2}, & \mathrm{if}\;1-r_{2}\leq \alpha <1-r_{1},\\ \frac{1-\alpha }{2r_{1}}p\Delta t-\left(|q\right|+\frac{1}{2}|f|)\Delta t^{2}, & \mathrm{if}\;1-r_{1}\leq \alpha <1, \end{array}\right.$$
(23)$${\left[p\eta +\left(|q\right|+\frac{1}{2}\left|f|\right)\eta^{2}\right]}_{\sup}\left(\alpha \right)\leq \left\{\begin{array}{cc} \left(1-\frac{\alpha }{2(1-r_{2})}\right)p\Delta t+\left(|q\right|+\frac{1}{2}|f|)\Delta t^{2}, & \mathrm{if}\;0<\alpha <1-r_{2},\\ \frac{1}{2}p\Delta t+\left(|q\right|+\frac{1}{2}|f|)\Delta t^{2}, & \mathrm{if}\;1-r_{2}\leq \alpha <1-r_{1},\\ \frac{1-\alpha }{2r_{1}}p\Delta t+\left(|q\right|+\frac{1}{2}|f|)\Delta t^{2}, & \mathrm{if}\;1-r_{1}\leq \alpha <1, \end{array}\right.$$(2)if p<0, then
(24)$${\left[p\eta -\left(|q\right|+\frac{1}{2}\left|f|\right)\eta^{2}\right]}_{\sup}\left(\alpha \right)\geq \left\{\begin{array}{cc} \frac{\alpha }{2r_{1}}p\Delta t-\left(|q\right|+\frac{1}{2}|f|)\Delta t^{2}, & \mathrm{if}\;0<\alpha \leq r_{1},\\ \frac{1}{2}p\Delta t-\left(|q\right|+\frac{1}{2}|f|)\Delta t^{2}, & \mathrm{if}\;r_{1}<\alpha \leq r_{2},\\ \left(1-\frac{1-\alpha }{2(1-r_{2})}\right)p\Delta t-\left(|q\right|+\frac{1}{2}|f|)\Delta t^{2}, & \mathrm{if}\;r_{2}<\alpha <r_{1}, \end{array}\right.$$
(25)$${\left[p\eta +\left(|q\right|+\frac{1}{2}\left|f|\right)\eta^{2}\right]}_{\sup}\left(\alpha \right)\leq \left\{\begin{array}{cc} \frac{\alpha }{2r_{1}}p\Delta t+\left(|q\right|+\frac{1}{2}|f|)\Delta t^{2}, & \mathrm{if}\;0<\alpha \leq r_{1},\\ \frac{1}{2}p\Delta t+\left(|q\right|+\frac{1}{2}|f|)\Delta t^{2}, & \mathrm{if}\;r_{1}<\alpha \leq r_{2},\\ \left(1-\frac{1-\alpha }{2(1-r_{2})}\right)p\Delta t+\left(|q\right|+\frac{1}{2}|f|)\Delta t^{2}, & \mathrm{if}\;r_{2}<\alpha <r_{1}, \end{array}\right.$$Therefore,
(1)if p≥0, then
(26)$${\left[Q\right]}_{\sup}\left(\alpha \right)\geq \left\{\begin{array}{cc} {}[\frac{\sqrt{3}}{\pi }\ln\frac{1-\alpha -\varepsilon }{\alpha +\varepsilon }\left|a\right|+\left(1-\frac{\alpha }{2(1-r_{2})}\right)p]\Delta t-h\Delta t^{2}, & \mathrm{if}\;0<\alpha <1-r_{2},\\ {}[\frac{\sqrt{3}}{\pi }\ln\frac{1-\alpha -\varepsilon }{\alpha +\varepsilon }\left|a\right|+\frac{1}{2}p]\Delta t-h\Delta t^{2}, & \mathrm{if}\;1-r_{2}\leq \alpha <1-r_{1},\\ {}[\frac{\sqrt{3}}{\pi }\ln\frac{1-\alpha -\varepsilon }{\alpha +\varepsilon }\left|a\right|+\frac{1-\alpha }{2r_{1}}p]\Delta t-h\Delta t^{2}, & \mathrm{if}\;1-r_{1}\leq \alpha <1, \end{array}\right.$$
(27)$${\left[Q\right]}_{\sup}\left(\alpha \right)\leq \left\{\begin{array}{cc} {}[\frac{\sqrt{3}}{\pi }\ln\frac{1-\alpha +\varepsilon }{\alpha -\varepsilon }\left|a\right|+\left(1-\frac{\alpha }{2(1-r_{2})}\right)p]\Delta t+h\Delta t^{2}, & \mathrm{if}\;0<\alpha <1-r_{2},\\ {}[\frac{\sqrt{3}}{\pi }\ln\frac{1-\alpha +\varepsilon }{\alpha -\varepsilon }\left|a\right|+\frac{1}{2}p]\Delta t+h\Delta t^{2}, & \mathrm{if}\;1-r_{2}\leq \alpha <1-r_{1},\\ {}[\frac{\sqrt{3}}{\pi }\ln\frac{1-\alpha +\varepsilon }{\alpha -\varepsilon }\left|a\right|+\frac{1-\alpha }{2r_{1}}p]\Delta t+h\Delta t^{2}, & \mathrm{if}\;1-r_{1}\leq \alpha <1, \end{array}\right.$$(2)if p<0, then
(28)$${\left[Q\right]}_{\sup}\left(\alpha \right)\geq \left\{\begin{array}{cc} {}[\frac{\sqrt{3}}{\pi }\ln\frac{1-\alpha -\varepsilon }{\alpha +\varepsilon }\left|a\right|+\frac{\alpha }{2r_{1}}p]\Delta t-h\Delta t^{2}, & \mathrm{if}\;0<\alpha <1-r_{2},\\ {}[\frac{\sqrt{3}}{\pi }\ln\frac{1-\alpha -\varepsilon }{\alpha +\varepsilon }\left|a\right|+\frac{1}{2}p]\Delta t-h\Delta t^{2}, & \mathrm{if}\;1-r_{2}\leq \alpha <1-r_{1},\\ {}[\frac{\sqrt{3}}{\pi }\ln\frac{1-\alpha -\varepsilon }{\alpha +\varepsilon }\left|a\right|+\left(1-\frac{1-\alpha }{2(1-r_{2})}\right)p]\Delta t-h\Delta t^{2}, & \mathrm{if}\;1-r_{1}\leq \alpha <1, \end{array}\right.$$
(29)$${\left[Q\right]}_{\sup}\left(\alpha \right)\leq \left\{\begin{array}{cc} {}[\frac{\sqrt{3}}{\pi }\ln\frac{1-\alpha +\varepsilon }{\alpha -\varepsilon }\left|a\right|+\frac{\alpha }{2r_{1}}p]\Delta t+h\Delta t^{2}, & \mathrm{if}\;0<\alpha <1-r_{2},\\ {}[\frac{\sqrt{3}}{\pi }\ln\frac{1-\alpha +\varepsilon }{\alpha -\varepsilon }\left|a\right|+\frac{1}{2}p]\Delta t+h\Delta t^{2}, & \mathrm{if}\;1-r_{2}\leq \alpha <1-r_{1},\\ {}[\frac{\sqrt{3}}{\pi }\ln\frac{1-\alpha +\varepsilon }{\alpha -\varepsilon }\left|a\right|+\left(1-\frac{1-\alpha }{2(1-r_{2})}\right)p]\Delta t+h\Delta t^{2}, & \mathrm{if}\;1-r_{1}\leq \alpha <1, \end{array}\right.$$where $h=[{\left(\frac{\sqrt{3}}{\pi }\ln\frac{2-\varepsilon }{\varepsilon }\right)}^{2}\left(|b\right|+\frac{1}{2}\left|f|\right)+\left|q\right|+\frac{1}{2}f]$.
(1)If p≥0, then by (Equation 17), (Equation 27), for Δt>0, if $0<\alpha <1-r_{2}$, there exists a control $u=u_{\varepsilon ,\Delta t}$ such that
(30)$$\begin{array}{cc} -\varepsilon \Delta t & \leq f\Delta t+J_{t}\Delta t+\nu J_{x}\Delta t+m\left(L_{t}\right)J_{l}\Delta t+{\left[Q\right]}_{\sup}\left(\alpha \right)+o\left(\Delta t\right)\\ \; & \leq f\Delta t+J_{t}\Delta t+\nu J_{x}\Delta t+m\left(L_{t}\right)J_{l}\Delta t+\frac{\sqrt{3}}{\pi }\ln\frac{1-\alpha +\varepsilon }{\alpha -\varepsilon }\left|a\right|\Delta t\\ \; & +\left(1-\frac{\alpha }{2(1-r_{2})}\right)p\Delta t+h\Delta t^{2}+o\left(\Delta t\right). \end{array}$$Since
$$\begin{array}{cc} \; & a=\lambda \left[J_{x}+\nu J_{xx}\Delta t+J_{tx}\Delta t+m\left(L_{t}\right)J_{xl}\Delta t\right]+n\left(L_{t}\right)\left[J_{l}+m\left(L_{t}\right)J_{ll}\Delta t+J_{tl}\Delta t+\nu J_{xl}\Delta t\right],\\ & p=\chi [J_{x}+\nu J_{xx}\Delta t+J_{tx}\Delta t+m\left(L_{t}\right)J_{xl}\Delta t]. \end{array}$$Substituting them into (Equation 30) and dividing both sides of the above inequality by Δt, we get
(31)$$\begin{array}{cc} -\varepsilon & \leq f+J_{t}+\nu J_{x}+m\left(L_{t}\right)J_{l}+\frac{\sqrt{3}}{\pi }\ln\frac{1-\alpha +\varepsilon }{\alpha -\varepsilon }|\lambda J_{x}+n\left(L_{t}\right)J_{l}+h_{1}\left(\Delta t\right)|\\ \; & +\left(1-\frac{\alpha }{2(1-r_{2})}\right)\left[\chi J_{x}+h_{2}\left(\Delta t\right)\right]+h_{3}\left(\varepsilon ,\Delta t\right)\\ \; & \leq J_{t}+\underset{D_{t}\in D}{\sup}\{f+\nu J_{x}+m\left(L_{t}\right)J_{l}+\frac{\sqrt{3}}{\pi }\ln\frac{1-\alpha +\varepsilon }{\alpha -\varepsilon }|\lambda J_{x}+n\left(L_{t}\right)J_{l}+h_{1}\left(\Delta t\right)|\\ \; & +\left(1-\frac{\alpha }{2(1-r_{2})}\right)\left[\chi J_{x}+h_{2}\left(\Delta t\right)\right]\}+h_{3}\left(\varepsilon ,\Delta t\right), \end{array}$$
where
$$\begin{array}{cc} \; & h_{1}\left(\Delta t\right)=\lambda \left(\nu J_{xx}\Delta t+J_{tx}\Delta t+m\left(L_{t}\right)J_{xl}\Delta t\right)+n\left(L_{t}\right)\left(m\left(L_{t}\right)J_{ll}\Delta t+J_{tl}\Delta t+\nu J_{xl}\Delta t\right),\\ \; & h_{2}\left(\Delta t\right)=\chi \left(\nu J_{xx}\Delta t+J_{tx}\Delta t+m\left(L_{t}\right)J_{xl}\Delta t\right),\\ \; & h_{3}\left(\varepsilon ,\Delta t\right)=h\Delta t, \end{array}$$
and $h_{1}\left(\Delta t\right)\to 0$, $h_{2}\left(\Delta t\right)\to 0$, $h_{3}\left(\varepsilon ,\Delta t\right)\to 0$ as Δt→0. Letting Δt→0 and then ε→0 results in
(32)$$\begin{array}{cc} 0\leq & J_{t}+\underset{D_{t}\in D}{\sup}\{f+\nu J_{x}+m\left(L_{t}\right)J_{l}+\frac{\sqrt{3}}{\pi }\ln\frac{1-\alpha }{\alpha }|\left(\lambda J_{x}+n\left(L_{t}\right)J_{l}\right)|\\ \; & +\left(1-\frac{\alpha }{2(1-r_{2})}\right)\chi J_{x}\}, \end{array}$$
if $\chi J_{x}\geq 0$ and $0<\alpha <1-r_{2}$.In the same way, by (Equation 17) and (Equation 26), we can get
(33)$$\begin{array}{cc} 0\geq & J_{t}+\underset{D_{t}\in D}{\sup}\{f+\nu J_{x}+m\left(L_{t}\right)J_{l}+\frac{\sqrt{3}}{\pi }\ln\frac{1-\alpha }{\alpha }|\left(\lambda J_{x}+n\left(L_{t}\right)J_{l}\right)|\\ \; & +\left(1-\frac{\alpha }{2(1-r_{2})}\right)\chi J_{x}\}, \end{array}$$
if $\chi J_{x}\geq 0$ and $0<\alpha <1-r_{2}$.Combining (Equation 32) and (Equation 33), we obtain
(34)$$-J_{t}=\underset{D_{t}\in D}{\sup}\left\{f+\nu J_{x}+m\left(L_{t}\right)J_{l}+\frac{\sqrt{3}}{\pi }\ln\frac{1-\alpha }{\alpha }|\left(\lambda J_{x}+n\left(L_{t}\right)J_{l}\right)|+\left(1-\frac{\alpha }{2(1-r_{2})}\right)\chi J_{x}\right\},$$
if $\chi J_{x}\geq 0$ and $0<\alpha <1-r_{2}$.According to (Equation 17), (Equation 26) and (Equation 27), using the similar techniques, we have
(35)$$-J_{t}=\underset{D_{t}\in D}{\sup}\left\{f+\nu J_{x}+m\left(L_{t}\right)J_{l}+\frac{\sqrt{3}}{\pi }\ln\frac{1-\alpha }{\alpha }|\left(\lambda J_{x}+n\left(L_{t}\right)J_{l}\right)|+\frac{1}{2}\chi J_{x}\right\},$$
if $\chi J_{x}\geq 0$, $1-r_{2}\leq \alpha <1-r_{1}$ and
(36)$$-J_{t}=\underset{D_{t}\in D}{\sup}\left\{f+\nu J_{x}+m\left(L_{t}\right)J_{l}+\frac{\sqrt{3}}{\pi }\ln\frac{1-\alpha }{\alpha }|\left(\lambda J_{x}+n\left(L_{t}\right)J_{l}\right)|+\frac{1-\alpha }{2r_{1}}\chi J_{x}\right\},$$
if $\chi J_{x}\geq 0$ and $1-r_{1}\leq \alpha <1$.
(2)If p<0, similar to the above method, we use (Equation 17), (Equation 29), (Equation 33) to derive the equation of optimality for $\chi J_{x}<0$. Therefore, Theorem 2 is proved. □

## 5. An Optimal Control Problem of DC Pension Fund

In recent years, pension fund management has become a popular and significant subject to retirees because it plays an essential role in the financial market and in the social security system. The dynamic control-theoretical framework was first applied to a pension fund problem by [4] by assuming that the pension fund can be invested in a risk-free asset and a risky asset whose return follows random jump-diffusion processes. Ref. [27] assumed risk asset returns follow an uncertain process with jump and made use of optimistic value criterion to optimize objective function of involving uncertain variables. We assume that the contribution of pension is related to the salary factors of members, then the DC pension plan control problem may be solved by the optimistic value model of uncertain optimal control with jump.

### 5.1. Finance Market

We assume that the financial market consists of two assets, a risk-free asset (i.e., the bank account or bond), and a single risk asset (i.e., stock).

The price of the risk-free asset $S_{0}\left(t\right)$ at the time *t* evolves according to the following uncertain process
(37)$$\frac{dS_{0}\left(t\right)}{S_{0}\left(t\right)}=rdt,$$
where *r* is a constant and represents the risk-free interest rate.

The price of the risk asset S(t) at the time *t* evolves according to the following uncertain process with jump
(38)$$\frac{dS(t)}{S(t)}=\mu dt+\sigma_{1}dC\left(t\right)+\sigma_{2}dV\left(t\right),$$
where μ is the appreciation rate of the risk asset and $\sigma_{1},\sigma_{2}$ is the volatility rate, μ, $\sigma_{1}$, and $\sigma_{2}$ are all positive constants, and C(t) is a canonical process, V(t) is a *V*-jump uncertain process. In general, we assume that μ>r.

The salary level is denoted by L(t) at time *t*. We assume that L(t) follows a uncertain growth given by
(39)$$\frac{dL(t)}{L(t)}=\mu_{L}dt+\sigma_{L}dC\left(t\right),$$
where $\mu_{L}$ is the expected rate of return on salary, $\sigma_{L}$ is the salary volatility caused by the fluctuation of risk asset.

We assume that the pension contribution rate is θL(t), where 0<θ<1 is a constant.

### 5.2. Wealth Process

Assuming that the plan managers can invest in both the risk-free and the risky assets described by (Equation 37) and (Equation 38), respectively, and use the fund to pay retirement benefits. Let $x_{0}$ denote the initial wealth of this fund, ω(t) denote the investment proportion that the plan managers invest in the risky asset at time *t*, and X(t) denote the wealth of the pension fund at time *t* after adapting the investment strategy ω(t), B(t) is the pension payment rate at time *t*. Then the fund’s value follows the dynamics
(40)$$\left\{\begin{array}{c} \begin{array}{cc} dX(t) & =\left.[1-\omega (t)\right.]X\left(t\right)\frac{dS_{0}\left(t\right)}{S_{0}\left(t\right)}+\omega \left(t\right)X\left(t\right)\frac{dS(t)}{S(t)}+\theta L\left(t\right)\left[\mu_{L}dt+\sigma_{L}dC\left(t\right)\right]-B\left(t\right)dt,\\ X(0) & =x_{0}. \end{array} \end{array}\right.$$

Using (Equation 37)–(Equation 39), we can easily rewrite (Equation 40) as
(41)$$\left\{\begin{array}{c} \begin{array}{cc} dX(t) & =\left[X\left(t\right)r+\left(\mu -r\right)\omega \left(t\right)X\left(t\right)+\theta \mu_{L}L\left(t\right)-B\left(t\right)\right]dt+\left[\sigma_{1}\omega \left(t\right)X\left(t\right)+\sigma_{L}\theta L\left(t\right)\right]dC\left(t\right)\\ \; & +[\omega \left(t\right)X\left(t\right)\sigma_{2}]dV\left(t\right),\\ X_{0} & =x_{0}. \end{array} \end{array}\right.$$

### 5.3. Optimization Model

Our goal is to seek the optimal investment strategy ω(t) and payment rate B(t) to minimize the accumulated losses, thus we establish the following optimal model of pension fund.
(42)$$\left\{\begin{array}{c} \begin{array}{cc} \; & J\left(t,x,l\right)=\min_{\omega (t),B(t)}{\left\{\int_{t}^{\infty }e^{-\rho s}\left[\alpha_{1}{\left(B\left(s\right)-b_{m}\right)}^{2}+\alpha_{2}{\left(\omega \left(s\right)X\left(s\right)-x_{p}\right)}^{2}\right]ds\right\}}_{\inf}\left(\alpha \right)\\ \; & \mathrm{subject}\;\mathrm{to}\\ \; & dX\left(t\right)=\left[X\left(t\right)r+\left(\mu -r\right)\omega \left(t\right)X\left(t\right)+\theta \mu_{L}L\left(t\right)-B\left(t\right)\right]dt+\left[\sigma_{1}\omega \left(t\right)X\left(t\right)+\sigma_{L}\theta L\left(t\right)\right]dC\left(t\right)\\ \; & \;+[\omega \left(t\right)X\left(t\right)\sigma_{2}]dV\left(t\right),\\ \; & dL\left(t\right)=L\left(t\right)\left[\mu_{L}dt+\sigma_{L}dC\left(t\right)\right],\;X_{t}=x,\;L\left(t\right)=l, \end{array} \end{array}\right.$$
where, $\alpha_{1}>0$, $\alpha_{2}>0$, and $\alpha_{1}+\alpha_{2}=1$. α∈(0,1) denotes a given confidence level, ρ>0 denotes the discount rate. $b_{m}$ denotes the constant target contribution rate and $x_{p}$ denotes the constant target funding level.

It follows from Lemma 2 that model (Equation 42) is equivalent to the following model (Equation 43).
(43)$$\left\{\begin{array}{c} \begin{array}{cc} \; & J\left(t,x,l\right)=\max_{\omega (t),B(t)}{\left\{\int_{t}^{\infty }-e^{-\rho s}\left[\alpha_{1}{\left(B\left(s\right)-b_{m}\right)}^{2}+\alpha_{2}{\left(\omega \left(s\right)X\left(s\right)-x_{p}\right)}^{2}\right]ds\right\}}_{\sup}\left(\alpha \right)\\ \; & \mathrm{subject}\;\mathrm{to}\\ \; & dX\left(t\right)=\left[X\left(t\right)r+\left(\mu -r\right)\omega \left(t\right)X\left(t\right)+\theta \mu_{L}L\left(t\right)-B\left(t\right)\right]dt+\left[\sigma_{1}\omega \left(t\right)X\left(t\right)+\sigma_{L}\theta L\left(t\right)\right]dC\left(t\right)\\ \; & \;+[\omega \left(t\right)X\left(t\right)\sigma_{2}]dV\left(t\right),\\ \; & dL\left(t\right)=L\left(t\right)\left[\mu_{L}dt+\sigma_{L}dC\left(t\right)\right],\;X_{t}=x,\;L\left(t\right)=l. \end{array} \end{array}\right.$$

### 5.4. The Solution to the Model

By applying the equation of optimality (Equation 11), we get
(44)$$\begin{array}{cc} -J_{t} & =\max_{\omega (t),B(t)}\{-e^{-\rho t}\left[\alpha_{1}{\left(B\left(t\right)-b_{m}\right)}^{2}+\alpha_{2}{\left(\omega \left(t\right)x-x_{p}\right)}^{2}\right]+\left[xr+\left(\mu -r\right)\omega \left(t\right)x+\theta \mu_{L}l-B\left(t\right)\right]J_{x}\\ \; & +\mu_{L}lJ_{l}+\frac{\sqrt{3}}{\pi }\ln\frac{1-\alpha }{\alpha }|\left[\sigma_{1}\omega \left(t\right)x+\sigma_{L}\theta l\right]J_{x}+\sigma_{L}lJ_{l}|+k\omega \left(t\right)x\sigma_{2}J_{x}\}\\ \; & =\max_{\omega (t),B(t)}H\left(\omega \left(t\right),B\left(t\right)\right), \end{array}$$
where H(ω(t),B(t)) represents the term in the brackets.

Now we solve the (Equation 44)
(1)If $\left(\sigma_{1}\omega \left(t\right)x+\sigma_{L}\theta l\right)J_{x}+\sigma_{L}lJ_{l}\geq 0$, we differentiate the expression in brackets with respect to ω(t) and B(t) to find that
(45)$$\begin{array}{cc} \frac{\partial H(\omega (t),B(t))}{\partial \omega (t)}\; & =-2e^{-\rho t}\alpha_{2}\left(\omega \left(t\right)x-x_{p}\right)x+\left(\mu -r\right)xJ_{x}+\frac{\sqrt{3}}{\pi }\ln\frac{1-\alpha }{\alpha }\sigma_{1}xJ_{x}+kx\sigma_{2}J_{x}\\ \; & =0, \end{array}$$
(46)$$\frac{\partial H(\omega (t),B(t))}{\partial B(t)}=-2e^{-\rho t}\alpha_{1}\left(B\left(t\right)-b_{m}\right)-J_{x}=0.$$Solving Equations (Equation 45) and (Equation 46), we get
$$\omega \left(t\right)=\frac{1}{x}\left[x_{p}+\frac{1}{2\alpha_{2}}\tilde{k}e^{\rho t}J_{x}\right],\;B\left(t\right)=b_{m}-\frac{1}{2\alpha_{1}}e^{\rho t}J_{x},$$
where $\tilde{k}=\mu -r+\frac{\sqrt{3}}{\pi }\ln\frac{1-\alpha }{\alpha }\sigma_{1}+k\sigma_{2}$.Substituting them into (Equation 44) implies
(47)$$-J_{t}=\tilde{a}e^{\rho t}J_{x}^{2}+\left(xr+\theta \tilde{c}l+\tilde{b}\right)J_{x}+\tilde{c}lJ_{l},$$
where $\tilde{a}=\frac{1}{4\alpha_{1}}+\frac{1}{4\alpha_{2}}{\tilde{k}}^{2}$, $\tilde{b}=\tilde{k}x_{p}-b_{m}$, $\tilde{c}=\mu_{L}+\frac{\sqrt{3}}{\pi }\ln\frac{1-\alpha }{\alpha }\sigma_{L}$.Multiplying both sides of equation by $e^{\rho t}$
(48)$$-e^{\rho t}J_{t}=\tilde{a}{\left(e^{\rho t}J_{x}\right)}^{2}+\left(xr+\theta \tilde{c}l+\tilde{b}\right)e^{\rho t}J_{x}+\tilde{c}le^{\rho t}J_{l}.$$Next we solve the partial differential Equation (Equation 48).Supposing $J\left(t,x,l\right)=e^{-\rho t}Q\left(x,l\right)$, then differentiating both sides with respect to *t*, *x*, and *l*, then $J_{t}=-\rho e^{-\rho t}Q\left(x,l\right)$, $J_{x}=e^{-\rho t}Q_{x}$, $J_{l}=e^{-\rho t}Q_{l}$. Substituting them into (Equation 48) yields
(49)$$\rho Q\left(x,l\right)=\tilde{a}Q_{x}^{2}+\left(xr+\theta \tilde{c}l+\tilde{b}\right)Q_{x}+\tilde{c}lQ_{l}.$$Assuming $Q\left(x,l\right)=A\left(x^{2}+2hxl+bl^{2}+ax+gl+u\right)$, then $Q_{x}=2Ax+2Ahl+Aa$, $Q_{l}=2Abl+2Ahx+Ag)$. Substituting them into (Equation 49) yields
(50)$$\begin{array}{cc} \; & \left(\rho A-4A^{2}\tilde{a}-2Ar\right)x^{2}+\left(\rho Aa-4A^{2}a\tilde{a}-Aar-2A\tilde{b}\right)x\\ \; & +\left(\rho Ab-4A^{2}h^{2}\tilde{a}-2Ah\theta \tilde{c}-2Ab\tilde{c}\right)l^{2}+\left(\rho Ag-4A^{2}ha\tilde{a}-Aa\theta \tilde{c}-2Ah\tilde{b}-Ag\tilde{c}\right)l\\ \; & +\left(2\rho Ah-8A^{2}h\tilde{a}-2Ahr-2A\theta \tilde{c}-2Ah\tilde{c}\right)xl+\left(\rho Au-A^{2}a^{2}\tilde{a}-Aa\tilde{b}\right)=0. \end{array}$$Decomposing Equation (Equation 50) obtains
(51)$$\begin{array}{cc} \; & \rho A-4A^{2}\tilde{a}-2Ar=0,\;\rho Aa-4A^{2}a\tilde{a}-Aar-2A\tilde{b}=0,\\ \; & \rho Ab-4A^{2}h^{2}\tilde{a}-2Ah\theta \tilde{c}-2Ab\tilde{c}=0,\;\rho Ag-4A^{2}ha\tilde{a}-Aa\theta \tilde{c}-2Ah\tilde{b}-Ag\tilde{c}=0,\\ \; & 2\rho Ah-8A^{2}h\tilde{a}-2Ahr-2A\theta \tilde{c}-2Ah\tilde{c}=0,\;\rho Au-A^{2}a^{2}\tilde{a}-Aa\tilde{b}=0. \end{array}$$By solving Equation (Equation 51), we get
(52)$$A=\frac{\rho -2r}{4\tilde{a}},\;a=\frac{2\tilde{b}}{r},\;h=\frac{\theta \tilde{c}}{r-\tilde{c}},\;b={\left(\frac{\theta \tilde{c}}{r-\tilde{c}}\right)}^{2},\;g=\frac{2\theta \tilde{b}\tilde{c}}{r(r-\tilde{c})},\;u=\frac{{\tilde{b}}^{2}}{r^{2}}.$$Thus,
(53)$$\begin{array}{cc} J_{x} & =e^{-\rho t}Q_{x},\\ Q_{x} & =2Ax+2Ahl+Aa\\ \; & =\frac{\left(\rho -2r\right)}{2\tilde{a}}x+\frac{\theta \tilde{c}\left(\rho -2r\right)}{2\tilde{a}\left(r-\tilde{c}\right)}l+\frac{\tilde{b}\left(\rho -2r\right)}{2\tilde{a}r}. \end{array}$$Then,
(54)$$J_{x}=e^{-\rho t}\left[\frac{\left(\rho -2r\right)}{2\tilde{a}}x+\frac{\theta \tilde{c}\left(\rho -2r\right)}{2\tilde{a}\left(r-\tilde{c}\right)}l+\frac{\tilde{b}\left(\rho -2r\right)}{2\tilde{a}r}\right].$$So the optimal investment rate and the payment rate are determined, respectively, by
(55)$$\begin{array}{cc} \omega^{*}\left(t\right) & =\frac{1}{x}\left[x_{p}+\frac{1}{2\alpha_{2}}\tilde{k}e^{\rho t}J_{x}\right]\\ \; & =\frac{1}{x}\left[x_{p}+\frac{1}{2\alpha_{2}}\tilde{k}\left(\frac{\left(\rho -2r\right)}{2\tilde{a}}x+\frac{\theta \tilde{c}\left(\rho -2r\right)}{2\tilde{a}\left(r-\tilde{c}\right)}l+\frac{\tilde{b}\left(\rho -2r\right)}{2\tilde{a}r}\right)\right], \end{array}$$
(56)$$\begin{array}{cc} B^{*}\left(t\right) & =b_{m}-\frac{1}{2\alpha_{1}}e^{\rho t}J_{x}\\ \; & =b_{m}-\frac{1}{2\alpha_{1}}\left[\frac{\left(\rho -2r\right)}{2\tilde{a}}x+\frac{\theta \tilde{c}\left(\rho -2r\right)}{2\tilde{a}\left(r-\tilde{c}\right)}l+\frac{\tilde{b}\left(\rho -2r\right)}{2\tilde{a}r}\right]. \end{array}$$(2)If $\left(\sigma_{1}\omega \left(t\right)x+\sigma_{L}\theta l\right)J_{x}+\sigma_{L}lJ_{l}<0$, then applying the similar method to the above processes, we can get results (Equation 55) and (Equation 56), where $\tilde{k}=\mu -r-\frac{\sqrt{3}}{\pi }\ln\frac{1-\alpha }{\alpha }\sigma_{1}+k\sigma_{2}$.

**Theorem** **3.**
*For the optimization model (Equation 43), the optimal investment strategy is given by*

$$\omega^{*}\left(t\right)=\frac{1}{x}\left[x_{p}+\frac{1}{2\alpha_{2}}\tilde{k}\left(\frac{\left(\rho -2r\right)}{2\tilde{a}}x+\frac{\theta \tilde{c}\left(\rho -2r\right)}{2\tilde{a}\left(r-\tilde{c}\right)}l+\frac{\tilde{b}\left(\rho -2r\right)}{2\tilde{a}r}\right)\right],$$

*the payment rate is given by*

$$B^{*}\left(t\right)=b_{m}-\frac{1}{2\alpha_{1}}\left[\frac{\left(\rho -2r\right)}{2\tilde{a}}x+\frac{\theta \tilde{c}\left(\rho -2r\right)}{2\tilde{a}\left(r-\tilde{c}\right)}l+\frac{\tilde{b}\left(\rho -2r\right)}{2\tilde{a}r}\right],$$

*where*

$$\tilde{a}=\frac{1}{4\alpha_{1}}+\frac{1}{4\alpha_{2}}{\tilde{k}}^{2},\;\tilde{b}=\tilde{k}x_{p}-b_{m},\;\tilde{c}=\mu_{L}+\frac{\sqrt{3}}{\pi }\ln\frac{1-\alpha }{\alpha }\sigma_{L}.$$

*If $\left(\sigma_{1}\omega \left(t\right)x+\sigma_{L}\theta l\right)J_{x}+\sigma_{L}lJ_{l}\geq 0$,*

$$\tilde{k}=\mu -r+\frac{\sqrt{3}}{\pi }\ln\frac{1-\alpha }{\alpha }\sigma_{1}+k\sigma_{2},$$

*if $\left(\sigma_{1}\omega \left(t\right)x+\sigma_{L}\theta l\right)J_{x}+\sigma_{L}lJ_{l}<0$,*

$$\tilde{k}=\mu -r-\frac{\sqrt{3}}{\pi }\ln\frac{1-\alpha }{\alpha }\sigma_{1}+k\sigma_{2}.$$



**Remark** **3.**
*The optimal payment rate $B^{*}\left(t\right)$, the optimal investment proportion $\omega^{*}\left(t\right)$ and the optimal value J(t,x,l) depend on the parameters $x_{p},b_{m},\alpha_{1},\alpha_{2},\rho ,r,\theta ,l,\mu ,\mu_{L},\sigma_{1},\sigma_{2},\sigma_{L}$ and the fund amount x.*


## 6. Numerical Analysis

In this section, we provide a numerical analysis to characterize the dynamic behavior of the optimal investment strategy and the optimal payment rate. We fix the parameter values according to the modeling background. r=0.02, μ=0.05, $\sigma_{1}=0.15$, $\sigma_{2}=0.2$, $\mu_{L}=0.03$, $\sigma_{L}=0.06$, θ=0.02, α=0.95, $\alpha_{1}=0.6$, $\alpha_{2}=1-\alpha_{1}$, $x_{0}=0$, $x_{p}=5$, $p_{m}=5$, ρ=0.05, $r_{1}=0.2$, $r_{2}=0.7$, when α=0.2, $k=1-\alpha /(2\left(1-r_{2}\right))$, when α=0.5, k=0.5, when α=0.8, $k=\left(1-\alpha \right)/\left(2r_{1}\right)$.

Figure 1 shows the effect of parameter α on the optimal investment proportion $\omega^{*}\left(t\right)$ and the optimal payment rate $B^{*}\left(t\right)$. From Figure 1a,b, we can see that both $\omega^{*}\left(t\right)$ and $B^{*}\left(t\right)$ increase when α increases, with all other parameters being fixed. The confidence level α reflects the risk preference of a pension fund manager. Larger α means that the pension fund manager is risk averse. In other words, the pension fund manager would like more prudently to run the fund to achieve his/ her expected management targets. Figure 1 says that for a prudent manager, he/she can invest more in the financial market so that can make more profit, while he/she can also pay more money to the retirees.

Effects of model main parameters μ, $\sigma_{1}$ and $\sigma_{2}$ on $\omega^{*}\left(t\right)$ and $B^{*}\left(t\right)$ are shown in Figure 2. The graphs in Figure 2a–f plot the values of the optimal investment proportion $\omega^{*}\left(t\right)$ and the optimal payment rate $B^{*}\left(t\right)$ with respect to the wealth *x* at time 0, when the parameters μ, $\sigma_{1}$ and $\sigma_{2}$ influencing the stock’s price change. From Figure 2a,b, we can find that the values of $\omega^{*}\left(t\right)$ and $B^{*}\left(t\right)$ increase as μ increases. This is consistent with the intuition that when the return rate μ of the stock becomes higher, the pension fund manager would naturally like to invest more in the stock to make more profit. At the same time, the pension fund manager is able to pay more to the retirees who participate in the plan. From Figure 2c–f, we can see that $\omega^{*}\left(t\right)$ and $B^{*}\left(t\right)$ decrease as $\sigma_{1}$ and $\sigma_{2}$ increases, respectively. The corresponding economic explanation is as follows. Higher values of $\sigma_{1}$ and $\sigma_{2}$ represent higher uncertainty of the fluctuation of the price of the stock. In other words, the higher the values of $\sigma_{1}$ and $\sigma_{2}$ are, the more risky the stock is. Therefore, a risk averse pension fund manager would most likely to reduce the amount invested in the stock, and has to lower the payment rate to the pension members because of the reduction of the profit from the stock market.

In Figure 3, we examine the sensitivity of $\omega^{*}\left(t\right)$ and $B^{*}\left(t\right)$ with respect to the parameters influencing salary level. From Figure 3a–d, we can see that both $\omega^{*}\left(t\right)$ and $B^{*}\left(t\right)$ are insensitive to $\mu_{L}$ and $\sigma_{L}$.

## 7. Conclusions

In this paper, we have proposed a new optimal control model for uncertain systems. Unlike the classic optimal control model for uncertain systems, the proposed new optimal control model takes into account environmental factors. Under the optimistic value criterion, we have established the principle and equation of optimality. As an example, an application to DC pension plans has been given to illustrate the proposed optimal control model for uncertain systems. Numerical studies are also given to show the sensitivity analysis of the optimal solution to the model parameters. For further topics, it would be interesting to consider the principle and equation of optimality under other optimality criteria such as the Hurwitz criterion.

## Figures and Tables

**Figure 1 entropy-24-00734-f001:**
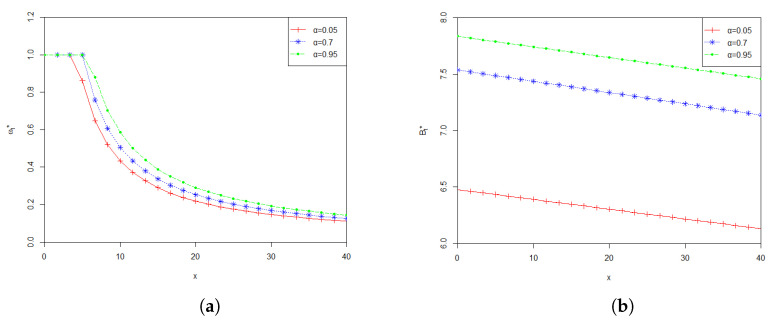
(**a**) Effect of α on the optimal investment proportion $\omega^{*}\left(t\right)$; (**b**) Effect of α on the optimal payment rate $B^{*}\left(t\right)$.

**Figure 2 entropy-24-00734-f002:**
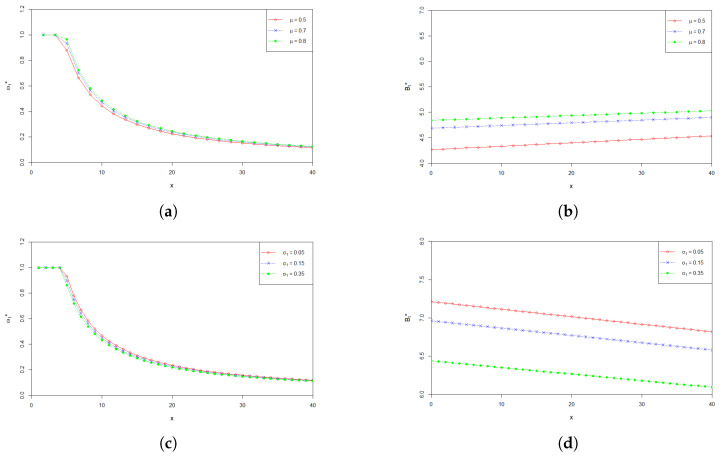
(**a**) Effect of μ on the optimal investment proportion $\omega^{*}\left(t\right)$; (**b**) Effect of μ on the optimal payment rate $B^{*}\left(t\right)$; (**c**) Effect of $\sigma_{1}$ on the optimal investment proportion $\omega^{*}\left(t\right)$; (**d**) Effect of $\sigma_{1}$ on the optimal payment rate $B^{*}\left(t\right)$; (**e**) Effect of $\sigma_{2}$ on the optimal investment proportion $\omega^{*}\left(t\right)$; (**f**) Effect of $\sigma_{2}$ on the optimal payment rate $B^{*}\left(t\right)$.

**Figure 3 entropy-24-00734-f003:**
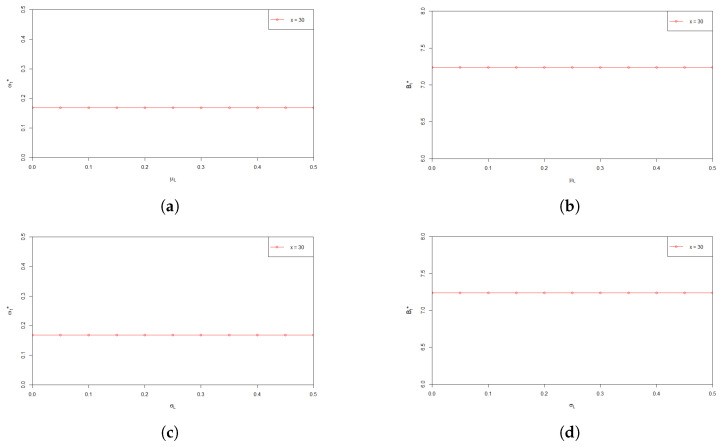
(**a**) Effect of $\mu_{L}$ on the optimal investment proportion $\omega^{*}\left(t\right)$; (**b**) Effect of $\mu_{L}$ the optimal payment rate $B^{*}\left(t\right)$; (**c**) Effect of $\sigma_{L}$ on the optimal investment proportion $\omega^{*}\left(t\right)$; (**d**) Effect of $\sigma_{L}$ the optimal payment rate $B^{*}\left(t\right)$.

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
