# Peer review of "Optimal Control of Background-Based Uncertain Systems with Applications in DC Pension Plan"

_entropy, 2022, doi:10.3390/e24050734_

Round 1

Reviewer 1 Report

see the file

Reviewer 2 Report

The article deals with a new optimal control model for uncertain systems with jump. Background-state variables are incorporated, where the background-state variables are governed by an uncertain differential equation. application in DC pension plans illustrates the effectiveness of the proposed model. The application of uncertainty theory with its applications in finance and insurance are practical example of using the proposed theory.

The introduction introduces and describes the problem of stochastic control with jump with related references link. Also, other results of relevant works are given in the introduction of the article.

Uncertainty theory is described in detail in the next chapter. The problem of uncertain optimal control is further addressed in this article. Equation of optimality is further addressed in the following section. An important part of the article is the practical application of the problem in the field of finance market. Optimization model and solution of the model is solved in the next part of article.

The main contribution of the article is a proposal and a new optimal control model for uncertain systems. The article is technically very well processed, and the methods used correspond to the nature of the problem

Comments:

The conclusion of the article is very brief. In the conclusion, the evaluation of the results and comparison with the results of other authors should be critical. In the conclusion, describe also what the fundamental own contribution and novelty of this article is. In the conclusion, there are also no plans for further research in this work.

Check references, as many are missing multiple bibliographic data (page numbers, edition, issue, volume number etc.). There are no references in this article to articles from this journal. If this article is in line with the focus of this journal, then it will not be a problem to find relevant references in this journal. Or then I recommend moving the article to another more appropriate journal.

The authors probably didn't see the template for this journal. This template was not used at all. Please rework this article into a template for this magazine.

Round 2

Reviewer 1 Report

See the file

Author Response

请参阅附件。
